# TGFß/activin-dependent activation of Torso controls the timing of the metamorphic transition in the red flour beetle *Tribolium castaneum*

**Sílvia Chafino, Roser Salvia⊙, Josefa Cruz⊙, David Martín⊙\*, Xavier Franch-Marro⊙\***

Institute of Evolutionary Biology (IBE, CSIC-Universitat Pompeu Fabra), Barcelona, Catalonia, Spain

\* david.martin@ibe.upf-csic.es (DM); xavier.franch@ibe.upf-csic.es (XF-M)

## Abstract

Understanding the mechanisms governing body size attainment during animal development is of paramount importance in biology. In insects, a crucial phase in determining body size occurs at the larva-pupa transition, marking the end of the larval growth period. Central to this process is the attainment of the threshold size (TS), a critical developmental checkpoint that must be reached before the larva can undergo metamorphosis. However, the intricate molecular mechanisms by which the TS orchestrates this transition remain poor understood. In this study, we investigate the role of the interaction between the Torso and TGFß/ activin signaling pathways in regulating metamorphic timing in the red flour beetle, *Tribolium castaneum*. Our results show that Torso signaling is required specifically during the last larval instar and that its activation is mediated not only by the prothoracicotropic hormone (Tc-Ptth) but also by Trunk (Tc-Trk), another ligand of the Tc-Torso receptor. Interestingly, we show that while Tc-Torso activation by Tc-Ptth determines the onset of metamorphosis, Tc-Trk promotes growth during the last larval stage. In addition, we found that the expression of *Tc-torso* correlates with the attainment of the TS and the decay of juvenile hormone (JH) levels, at the onset of the last larval instar. Notably, our data reveal that activation of TGFß/activin signaling pathway at the TS is responsible for repressing the JH synthesis and inducing *Tc-torso* expression, initiating metamorphosis. Altogether, these findings shed light on the pivotal involvement of the Ptth/Trunk/Torso and TGFß/activin signaling pathways as critical regulatory components orchestrating the TS-driven metamorphic initiation, offering valuable insights into the mechanisms underlying body size determination in insects.

## Author summary

Understanding the mechanisms that determine an animal's final body size is a fundamental question in biology. In animals, the majority of growth takes place during the juvenile stage, with adult size being established as they transition into adulthood. Hormones play a pivotal role in orchestrating this complex process. In the case of insects, the metamorphic

**Data Availability Statement:** All relevant data are within the manuscript and its Supporting Information files.

**Funding:** This project is supported by grants PGC2018-098427-B-I00 and PID2021-125661NB-I00 to D.M. and X.F-M. funded by MCIN/AEI/10.13039/501100011033 and by "ERDF A way of making Europe". The project is supported also by grants 2017 SGR 1030 and 2021 SGR 00417 to D.M. and X.F-M funded by the Departament de Recerca i Universitats de la Generalitat de Catalunya. S.C. was a recipient of a Juan de la Cierva contract FJC2019-041549-I funded by MCIN/AEI /10.13039/501100011033. The funders play no role in the study design, data collection and analysis, decision to publish, or preparation of the manuscript.

**Competing interests:** The authors have declared that no competing interests exist.

transition is induced by the interplay between the steroid hormone ecdysone and the sesquiterpenoid juvenile hormone (JH). Our research delved into the roles of the Torso and TGFß/Activin signaling pathways in regulating ecdysone and JH biosynthesis. Remarkably, we discovered that in contrast to other insects, the Torso pathway in *Tribolium* is activated by not one, but two ligands: the Prothoracicotropic hormone that regulates ecdysone production and Trunk that promotes growth in the last larval stage. Additionally, our investigations unveiled the dual functionality of the TGFß/Activin signaling pathway in the initiation of metamorphosis. Firstly, it lowers JH levels, setting in motion the genetic changes required for the metamorphic transition, including the upregulation of Torso and, secondly, facilitating ecdysone production. In summary, our research sheds light on the intricate regulatory network governing metamorphic timing and body size in insects.

## Introduction

How animals reach their final body size is a fundamental question in biology. Organism final body size depends on environmental cues as well as on the precise activation of genetic programs during development. In many animals, growth takes place mainly during the juvenile stage, and adult body size is therefore determined upon entering into adulthood [1]. Deciphering the molecular mechanisms underlying the timely decision to initiate adult maturation is, therefore, critical to understand how body size is controlled.

Hormones play an important role in the regulation of final body size, in part by coordinating the onset of adult maturation. For example, in holometabolous insects, whose growth period is restricted to a series of larval molts that accommodate the increasing size of the body, the onset of metamorphosis is triggered by a sharp increase of the steroid hormone ecdysone upon reaching a size-dependent developmental checkpoint [2]. Ecdysone is synthesized in a specialized organ named the prothoracic gland (PG) through the sequential catalytic action of a series of enzymes encoded by the *Halloween* gene family. These include the Rieske-domain protein *neverland* (*nvd*) [3,4], the short-chain dehydrogenase/reductase *shroud* (*sro*) [5] and the P450 enzymes *spook* (*spo*), *spookier* (*spok*), *phantom* (*phm*), *disembodied* (*dib*) and *shadow* (*sad*) [6–11]. The expression of the Halloween genes is highly regulated, being up-regulated in the PG at the metamorphic transition by the integrated activity of several receptor tyrosine kinases (RTKs) [12]. These RTKs include Torso [13–15], Epidermal Growth Factor Receptor (Egfr) [16,17], Anaplastic Lymphoma Kinase (Alk) and PDGF and VEGF receptor-related (Pvr) [12], all acting through the Ras/Raf/Erk MAP kinase signal transduction cascade, and the Insulin receptor (InR) acting through the PI3K/Akt pathway [18–21]. Among these RTKs, Torso is of particular interest since it acts as a key transducer of critical environmental cues such as nutrition status and population density [15,22]. In order to exert its regulatory function in the PG, Torso binds the Prothoracicotropic hormone (Ptth), a neuropeptide secreted from neuroendocrine cells located in the brain [13,23,24]. Although Ptth is considered the unique Torso ligand during postembryonic development, the fact that *torso* mutants showed a significant longer delay than *Ptth* null mutants [15], suggests the existence of additional ligands involved in the activation of Torso signaling. In fact, in addition to the Ptth, Torso possesses another ligand, Trunk (Trk), belonging to the cysteine knot growth factor superfamily, responsible for the activation of the pathway during the early embryo [14,25–27]. Although *trk* is not expressed during postembryonic stages of the fly *Drosophila melanogaster*, ectopic expression of a cleaved form of Trk in the PG is able to activate the pathway inducing precocious

pupariation [26]. Nevertheless, to date, the activation of the Torso pathway by Trk has only been observed in the embryogenesis of all the studied species [25,26].

Despite the well-documented role of the Ptth/Torso pathway in triggering the onset of metamorphosis in *Drosophila* and *Bombyx mori*, the regulation of the pathway itself is less understood. Whereas Ptth synthesis and release from the neuroendocrine brain cells seems to depend on nutritional and environmental cues [13,18,20,28], the regulatory mechanisms that control the expression of *torso* is poorly understood. In this regard, it has been shown in *Drosophila* that the activity of the Transforming Growth Factor ß (TGFß)/Activin pathway in the PG is required for the proper expression of *torso* [29]. However, since depletion of TGFß/activin affects cell growth and morphology [29], it is plausible that *torso* regulation by this pathway is a collateral rather than a direct effect. In addition, in the beetle *Tribolium* and hemimetabolous insects such as *Gryllus* and *Blattella*, TGFß/activin has also been described as a negative regulator of the biosynthesis of the anti-metamorphic juvenile hormone (JH) [30–32], suggesting a possible link between the decay of JH at the end of larval development and the activation of the Torso signaling pathway at the onset of metamorphosis.

Here, we use the red flour beetle *Tribolium* to study the regulation of metamorphic timing by the Torso signaling pathway. During development, the size of *Tribolium* increases over several larval instars until reaching a critical size-assessment checkpoint—the *threshold size* (TS)—that instructs the larva to enter metamorphosis at the ensuing molt, thus ending the growth period [33]. In laboratory conditions, the TS is reached at the onset of the seventh larval instar (L7), and is associated with the decay of JH and the consequence down-regulation of the anti-metamorphic transcription factor *Krüppel-homolg 1* (*Tc-Kr-h1*). As a result, the stage-specific transcription factors *Ecdysone inducible protein 93F* (*Tc-E93*) becomes up-regulated triggering metamorphosis [33]. Unfortunately, although the described genetic changes that control the nature of the metamorphic transition have been studied in detail [33,34], the molecular mechanisms underlying the regulation of the metamorphic timing in *Tribolium* remain to be clearly defined.

In the present study, we uncover the role and regulation of *Tc-torso* in the control of the metamorphic timing in *Tribolium*. We show that Tc-Torso is required specifically in the last larval instar of the beetle for the synthesis of ecdysone that promotes the metamorphic transition. Remarkably, we found that during this period Torso signaling is activated not only by Tc-Ptth but also by Tc-Trk. However, whereas Tc-Torso activation by Tc-Ptth determines the onset of metamorphosis, Tc-Trk promotes growth during the last larval stage. Moreover, our experiments indicate that *Tc-torso* expression depends on larvae reaching the TS at the onset of the last larval instar, and that the sustained increase in the expression levels of *Tc-torso* during the last larval instar depends on the decay of JH titers. Interestingly, we found that the activity of Baboon (Tc-Babo) and Myoglianin (Tc-Myo), two key components of the TGFß/activin signaling pathway, are required to repress JH synthesis and activate the TS-dependent induction of *Tc-torso* expression. Taken together, our results indicate that the Ptth/Trk/Torso and TGFß/activin pathways are critical component of the mechanism that controls the final body size of *Tribolium* by regulating growth and the timing of the metamorphic transition.

## Results

### Torso signaling regulates metamorphic timing in *Tribolium*

To investigate the role of Torso signaling in *Tribolium*, we first examined its temporal expression pattern during larval development. *Tc-torso* mRNA levels were low during early stages of larval development but strongly increased after entering into the last larval instar (L7), reaching the maximal level of expression at 24-48h, to decline thereafter until the end of the instar (Fig 1A). This result suggests that Torso signaling in *Tribolium* is necessary during the last

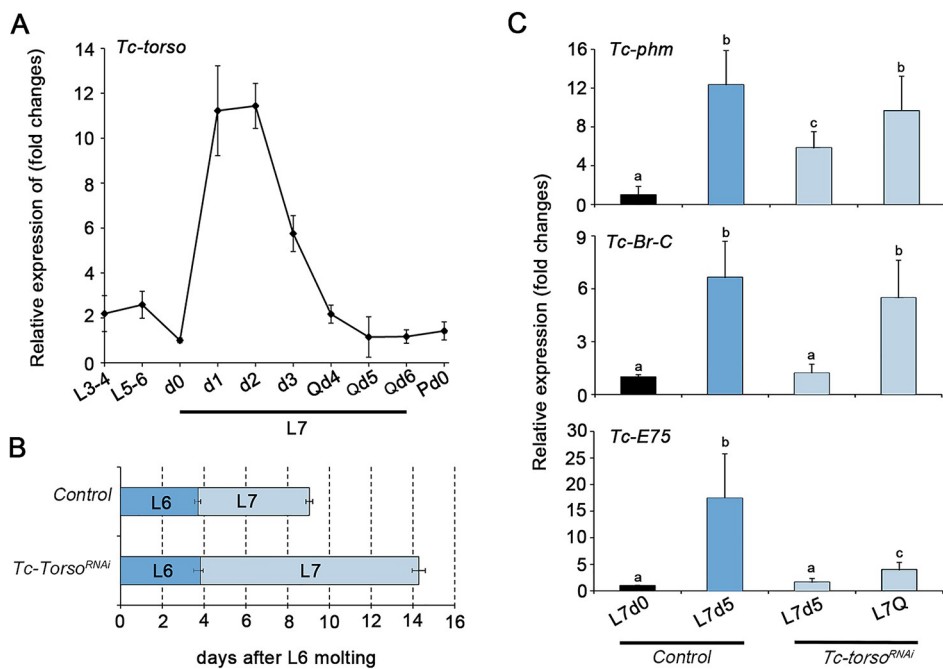

**Fig 1. Tc-Torso is a critical regulator of pupation in *Tribolium*.** (A) Temporal changes in *Tc-torso* mRNA levels measured by qRT-PCR from L3-4 instar larvae to pupa day 0 (Pd0). Transcript abundance values were normalized against the *Tc-Rpl32* transcript. Error bars indicate the standard error of the mean (SEM) (n = 5). (B) Developmental progression of newly molted L5 larvae after injection with either dsMock (*Control*) (n = 30) or *dsTc-torso* (n = 22). The bars represent the mean ± standard deviation (SD) for each developmental stage observed after the double-stranded RNA injection. (C) Transcript levels of *Tc-phm*, *Tc-Br-C*, and *Tc-E75* measured by qRT-PCR in 0 and 5-day-old L7 Control larvae and 5-day-old L7 (L7d5) and quiescent (L7Q) *Tc-torso^{RNAi}* larvae. Transcript abundance values were normalized against the *Tc-Rpl32* transcript. Error bars indicate the SEM (n = 3–5). Different letters represent groups with significant differences based on an ANOVA test (Tukey, p < 0.001). Raw data are in S1 Data (tab Fig 1).

larval stage of development. To test this possibility, we analysed the effects of blocking Torso signaling from early larval development by injecting dsRNA of *Tc-torso* in newly molted ante-penultimate L5 larvae (*Tc-torso^{RNAi}* animals). Specimens injected with dsMock were used as negative controls (*Control* animals). Under these conditions, *Tc-torso^{RNAi}* larvae molted with proper timing to L6 and then to L7, but showed a significant developmental delay in the pupation time (Fig 1B). These results confirmed that Torso signaling in *Tribolium* is specifically required during the last larval instar to control the timing of the metamorphic transition.

To determine whether the pupation delay observed in *Tc-torso^{RNAi}* larvae was caused by an ecdysone deficiency, we next analyzed the expression levels of the representative Halloween gene *Tc-phm*, as well as a number of well characterized ecdysone-dependent genes such as *Tc-Br-C* and *Tc-E75* that are used as proxies for ecdysone levels. Consistent with delayed pupation, *Tc-torso^{RNAi}* larvae presented a significant delay in the expression of all the analyzed genes when compared to *Control* animals (Fig 1C). Altogether, these results indicated that Torso activation during the last larval stage of *Tribolium* is required to control the production of ecdysone that timely triggers pupa formation.

## Tc-Ptth and Tc-Trk activate Torso signaling during the last larval instar

To further characterize the function of Torso during the metamorphic transition, we knocked-down this receptor specifically in the last larval stage. Under this treatment, *Tc-torso*-depleted

larva pupated with a delay of 7 days (Fig 2A). However, contrary to *Drosophila*, where deple-tion of either *Dm-Ptth* or *Dm-torso* resulted in bigger pupae [14], *Tc-torso*[RNAi] pupae were slightly smaller and lighter than *Control* pupae (Fig 2D and 2E). These results suggest the pos-sibility that postembryonic activation of Tc-Torso not only controls developmental timing through the regulation of ecdysone synthesis by Tc-Ptth activation, but also controls systemic growth rate. To further study this possibility, we analyzed the role of Ptth in Torso signaling activation in *Tribolium*. First, we measured the expression level of *Tc-Ptth* in last instar larvae, and found fluctuating levels with two peaks at day 2 and 4 (Fig 2B). Interestingly, depletion of *Tc-Ptth* in *Tribolium* (*Tc-Ptth*[RNAi] animals) induced a pupation delay of 5 days, compared to the almost 7 days observed in *Tc-torso*[RNAi] larvae (Fig 2A). Moreover, the absence of *Tc-Ptth* did not affect the weight of the resulting pupae when compared to *Control* pupae (Fig 2D and 2E). Taken together, the differences between *Tc-torso*[RNAi] and *Tc-Ptth*[RNAi] animals suggest that the activation of Torso signaling in *Tribolium* might not rely only on *Tc-Ptth* but also on additional ligands.

In *Drosophila*, the Torso pathway is also required for the formation of the most anterior and posterior regions of the embryo. During this process, Dm-Torso receptor is activated by the ligand Dm-Trk, which is synthetized in the early embryo [14,25–27]. Similarly, generation of abdominal segments in *Tribolium* requires the Tc-Trk-dependent activation of Torso sig-naling in the posterior region of the early embryo [26]. Although *Drosophila Dm-trk* is not expressed during larval development, ectopic expression of Dm-Trk induced a mild advance on pupariation [26], indicating that Dm-Trk is able to activate Torso signaling in the larval PG of *Drosophila*. Importantly, we were able to detect expression of *Tc-trk* in the last instar larvae of *Tribolium*, presenting a similar pattern than *Tc-torso*, with a peak of expression at day 3 (Fig 2C). Next, we injected dsRNA of *Tc-trunk* (*Tc-trk*[RNAi] animals) in L6 larvae to analyse its func-tional relevance during the last larval instar. In contrast to *Tc-Ptth*-depleted animals, *Tc-trk*[RNAi] individuals exhibited around 2 days of pupation delay (Fig 2A). Importantly, however, the resulting *Tc-trk*[RNAi] pupae were as small as the *Tc-torso*[RNAi] pupae (Fig 2D and 2E). These results suggest that the activation of Tc-Torso during the last stage of larval development depends on both Tc-Trunk and Tc-Ptth. To confirm this possibility, we knocked-down both ligands simultaneously (*Tc-Ptth*[RNAi] + *Tc-trk*[RNAi] animals). As expected, depletion of *Tc-trk* and *Tc-Ptth* phenocopied the absence of *Tc-torso*, as *Tc-trk*[RNAi] + *Tc-Ptth*[RNAi] animals pre-sented 7 days of delay and smaller pupae (Fig 2A, 2D and 2E). Altogether, these results show that both ligands, Tc-Trk and Tc-Ptth, act as Tc-Torso ligands to activate Torso signaling dur-ing the last larval stage in order to trigger a timely metamorphic transition.

## *Tc-trk* and *Tc-Ptth* exhibit tissue-specific expression patterns

Our results provide compelling evidence that Tc-Torso responds differently to its two ligands, Tc-Trk and Tc-Ptth. Depletion of *Tc-Ptth* leads to developmental delays, while silencing *Tc-trk* primarily results in a reduction in pupal size with a slight pupation delay. This observation suggests the potential for tissue-specific activation of *Tc-torso*. In this sense in *Drosophila*, Dm-Torso activation occurs during the final larval stage, affecting the PG and fat body, thereby reg-ulating ecdysone biosynthesis and animal growth, respectively [14,35]. To explore this possibil-ity in *Tribolium*, we quantified the mRNA levels of *Tc-torso* and its ligands, *Tc-trk* and *Tc-Ptth*, in different tissues. As anticipated, we observed high expression of *Tc-torso* in the head, likely the location of the PG, and also of *Tc-Ptth*, which aligns with previous reports of *Tc-Ptth* expression in a pair of neurons within the *Tribolium* brain (Fig 3A and 3B) [26]. This similar expression pattern suggests the activation of Tc-Torso by Tc-Ptth in the PG, regulating ecdy-sone biosynthesis. However, we also detected lower levels of *Tc-torso* in muscle, gut, and fat

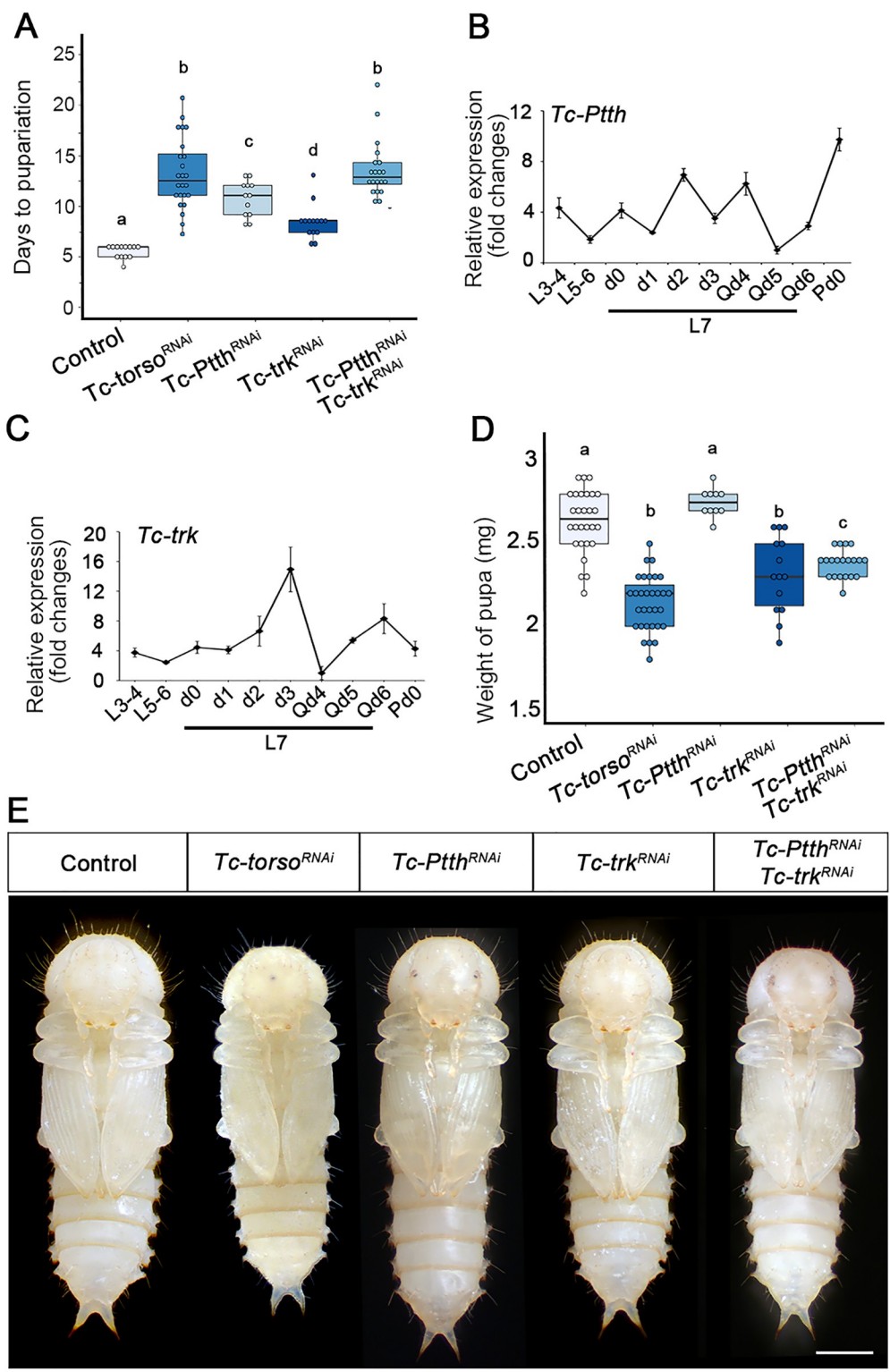

**Fig 2. Activation of Tc-Torso by Tc-Ptth and Tc-Trk during the final larval stage of *Tribolium*.** (A) Developmental duration of the last larval stage (L7) in newly molted L6 larvae injected with *dsMock* (Control) (n = 13), *dsTc-torso* (n = 33), *dsTc-Ptth* (n = 12), *dsTc-trk* (n = 14), and *dsTc-Ptth* + *dsTc-trk* (n = 16). (B-C) Temporal changes in *Tc-Ptth* mRNA (B) and *Tc-trk* mRNA (C) levels measured by qRT-PCR during larval development, from L3-4 to pupa day 0 (Pd0). Transcript abundance values are normalized against the *Tc-Rpl32* transcript. Error bars indicate the standard error

of the mean (SEM) (n = 5). (D) Body weight of the of *Control* (n = 10), *Tc-torso^RNAi^* (n = 10), *Tc-Ptth^RNAi^* (n = 12), *Tc-trk^RNAi^* (n = 14), and *Tc-Ptth^RNAi^ + dsTc-trk^RNAi^* (n = 16) larvae pupae. All weights were measured on day 0 of the pupal instar. (E) Ventral view of a *Control* and *Tc-torso^RNAi^*, *Tc-Ptth^RNAi^*, *Tc-trk^RNAi^*, and *Tc-Ptth^RNAi^ + Tc-trk^RNAi^* pupa. Scale bar represents 0.5 mm. In A and D, boxplots are used to represent the data, with black lines indicating medians, colored boxes showing the IQR, and bars indicating the upper and lower values. Values > ±1.5 x the IQR outside the box are considered outliers. Different letters represent groups with significant differences based on an ANOVA test (Tukey, $p < 0.001$). Raw data of A, B, C and D are in S1 Data (tab Fig 2).

body, hinting at the possible activation of Tc-Torso in these tissues (Fig 3A). In this sense, we found that *Tc-trk* expression was predominantly detected in the gut and fat body, implying a potential activation of the pathway by Tc-Trk in these tissues during the final larval stage (Fig 3C). Altogether, these findings imply the potential for distinct effects of Tc-Torso activation in different tissues mediated by its ligands, Tc-Ptth and Tc-Trk.

### *Tc-torso* expression is associated with the Threshold Size checkpoint

Our results above strongly suggest that Tc-Torso function is specifically of the last larval instar of *Tribolium*. It is at this stage that *Tribolium* larvae pass through a critical size-assessment checkpoint, the TS, which sets in motion the endocrine and genetic changes that trigger the metamorphic transition at the ensuing molt. The TS checkpoint in *Tribolium* is reached during the first 24 h after molting to the L7 stage and is associated with the stage-specific down-regulation of *Tc-Kr-h1* and up-regulation of *Tc-Br-C* and *Tc-E93* [33]. Since *Tc-torso* is strongly upregulated during the first 24 h of L7, we wondered whether this increase is also associated to the TS checkpoint. To address this issue, we starved L7 larvae before reaching the TS and measured mRNA levels of *Tc-torso* 72 h later. As Fig 4A shows, *Tc-torso* levels did not increase in larvae starved before the TS, confirming that the stage-specific up-regulation of *Tc-torso* is associated to larvae reaching the TS checkpoint.

Since attainment of the TS is also linked with the decline of JH levels [33], we next wondered if low levels of this hormone are required for the proper expression of *Tc-torso*. To study this, we treated L7 larvae with the JH-mimic methoprene at the TS to maintain high levels of this hormone throughout the larval stage. Under this condition, *Tc-torso* was properly up-regulated just after the TS but its expression significantly decreased thereafter when compared to the progressive increase observed in *Control* larvae (Fig 4B). Consequently, changing the identity of the last larval stage by the application of methoprene, reverted the delay induced by *Tc-torso^RNAi^* and induced the molting to a supernumerary L8 larva (Fig 4C). On the contrary, premature metamorphosis induced by depleting the rate limiting JH biosynthesis enzyme JH acid methyltransferase-3 (Tc-Jhamt) in newly emerged antepenultimate L4 instar larvae (*Tc-jhamt^RNAi^* animals) was delayed by *Tc-torso* depletion. Under these conditions, the majority of *Tc-jhamt^RNAi^* larvae molted to normal L5 larvae, and then to L6 underwent precocious metamorphosis after 5 days (Fig 4D). Interestingly, the time to pupation increased significantly when *Tc-jhamt* and *Tc-torso* were depleted simultaneously (*Tc-jhamt^RNAi^ + Tc-torso^RNAi^* animals) (Fig 4D), thus confirming that *Tc-torso* function does not depend on the number of larval stages the larva has been through but on whether the animal is in its last larval stage. Altogether, these results indicate that (1) the early upregulation of *Tc-torso* in L7 requires the larvae reaching the TS checkpoint; and (2) the ensuing increase in *Tc-torso* expression must occur in the presence of very low levels of JH.

### TGFß/Activin signaling pathway regulates *Tc-Torso* expression by repressing JH synthesis

Since the up-regulation of *Tc-torso* correlates with the decline in JH levels triggered by the TS checkpoint, we next wanted to study the relation between both processes. In this regard, it has

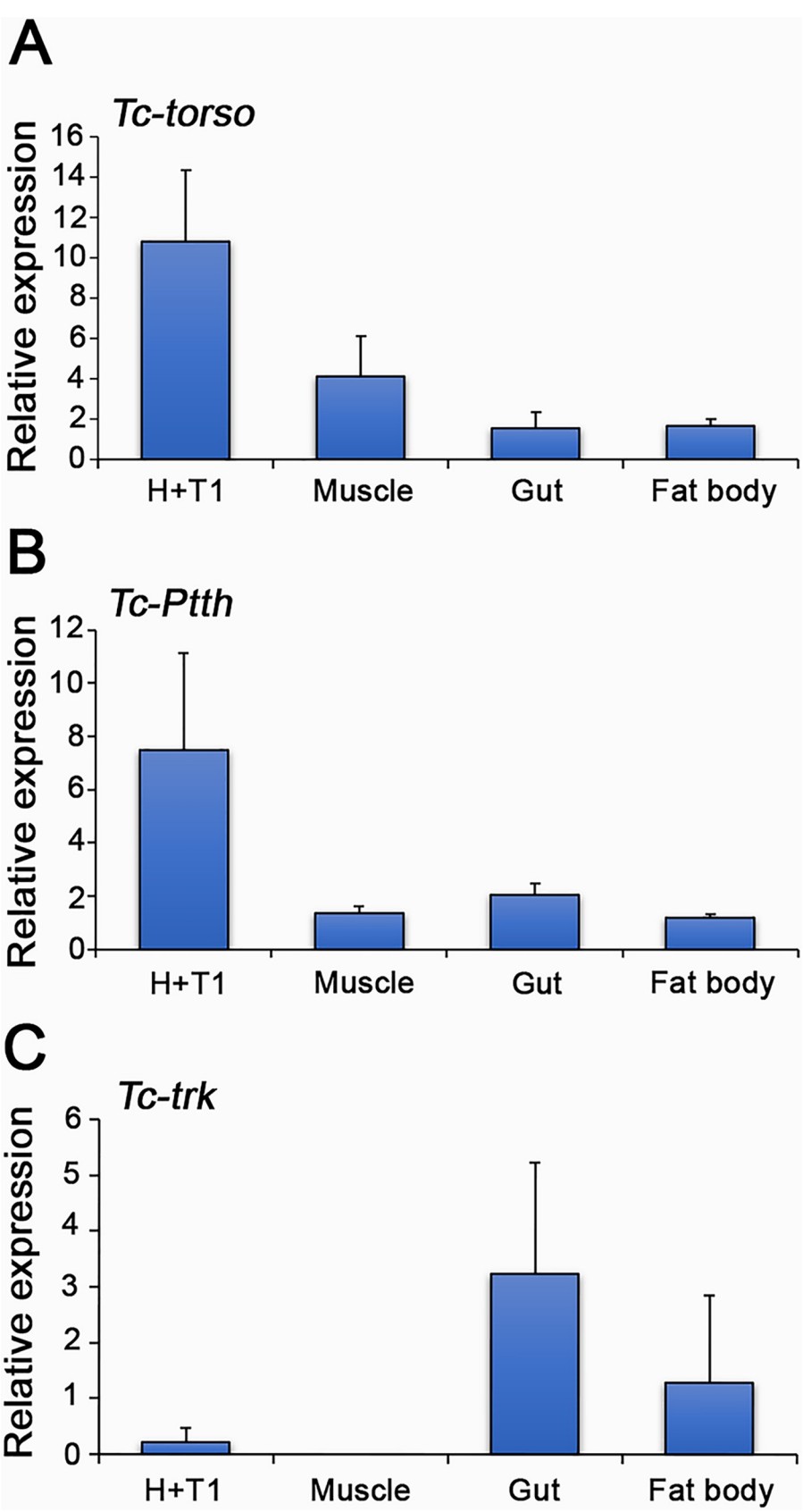

**Fig 3. Tissue specific expression of *Tc-torso*, *Tc-Ptth* and *Tc-trk*.** (A) *Tc-torso*, (B) *Tc-Ptth* and (C) *Tc-trk* mRNA levels measured by qRT-PCR in head and first thoracic segment (H+T1), muscle, gut and fat body of day 2 L7 larvae. For transcript analysis, equal amounts of total RNA were used. Error bars indicate the SEM (n = 5). Raw data are in S1 Data (tab Fig 3).

been recently shown that the TGFß/Activin signaling pathway is responsible for the decline of JH levels in a number of insect species [30–32]. We, therefore, wanted to ascertain whether the TGFß/Activin pathway regulates the expression of *Tc-torso* in the last larval instar. We first examined the expression of the Activin-like ligand Myoglianin (Tc-Myo) and its Type-I receptor Baboon (Tc-Babo) during the penultimate and last larval stages. Whereas *Tc-babo* mRNA levels persisted without major fluctuations through the last two larval instars, those of *Tc-myo* were more dynamic with a remarkable increase during the first two days and then oscillating throughout the rest of the instar (Fig 5A and 5B). Interestingly, while the expression of the

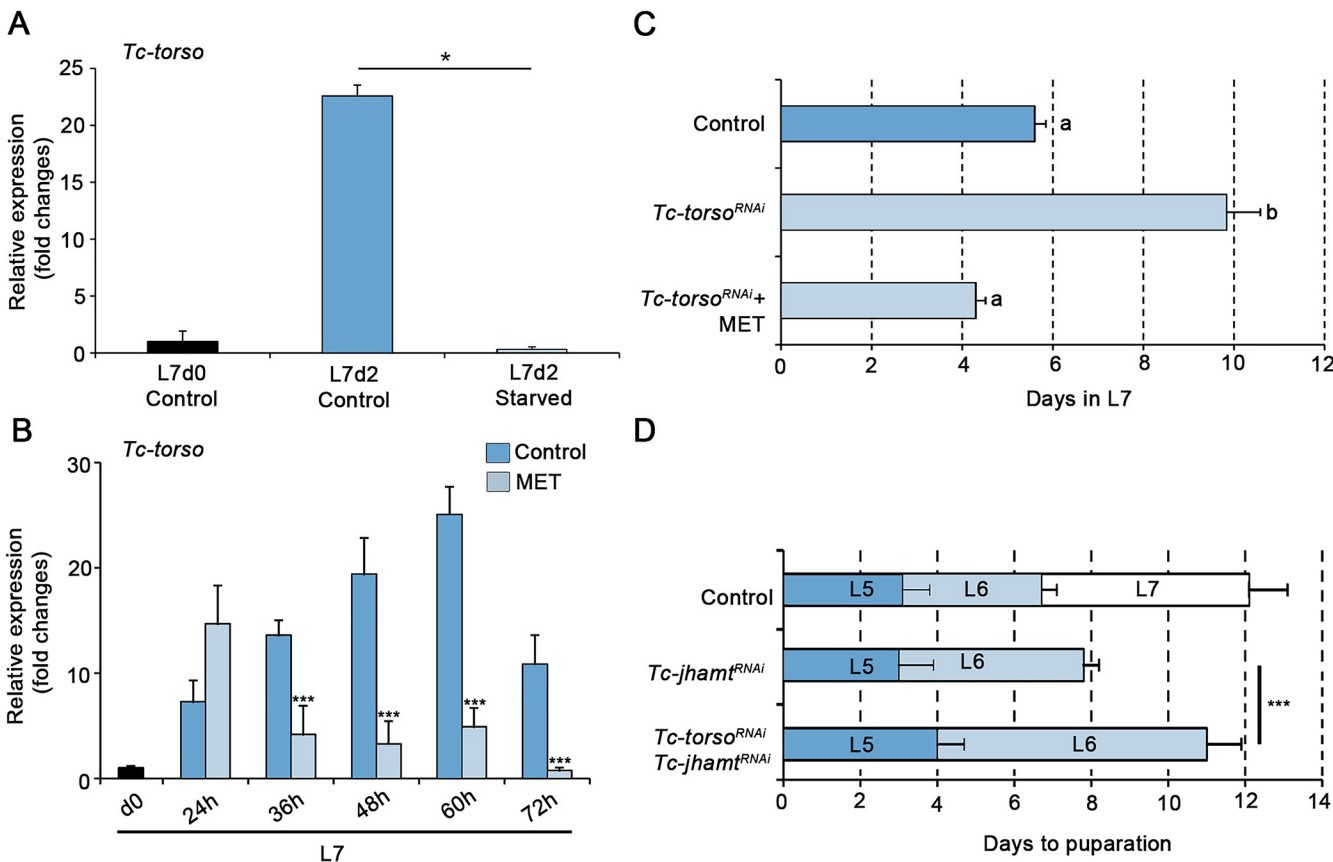

**Fig 4. Juvenile Hormone (JH) regulation of *Tc-torso* expression in *Tribolium*.** (A) Temporal changes in *Tc-torso* mRNA levels measured by qRT-PCR in *Control* and starved animals at the indicated stages. Transcript abundance values were normalized against the *Tc-RpL32* transcript. Average values of three independent datasets are shown with standard errors (n = 4–8). Asterisks indicate statistically significant differences at *p < 0.1 (t-test). (B) *Tc-torso* mRNA levels measured by qRT-PCR during the L7 larval instar of *Control* and methoprene-treated animals (MET). Note the dramatic decrease in *Tc-torso* transcription upon methoprene application. Transcript abundance values were normalized against the *Tc-RpL32* transcript. Error bars indicate the SEM (n = 5). Asterisks indicate statistically significant differences at ***p ≤ 0.001 (t-test). (C) Developmental duration of the last larval stage (L7) in newly molted L6 larvae injected with *dsMock* (*Control*) (n = 20), *dsTc-torso* (n = 23), and *dsTc-torso* + methoprene (n = 23). The developmental delay induced by depletion of *Tc-torso* is abolished by the application of methoprene. (D) Developmental progression of newly molted L4 larvae after injection with either *dsMock* (*Control*) (n = 15), *dsTc-jhamt* (n = 25) or *dsTc-torso* + *dsTc-jhamt* (n = 21). The bars represent the mean ± standard deviation (SD) for each developmental stage observed after the double-stranded RNA injection. Error bars indicate the SEM. Different letters represent groups with significant differences according to an ANOVA test (Tukey, p < 0.001). Raw data are in S1 Data (tab Fig 4).

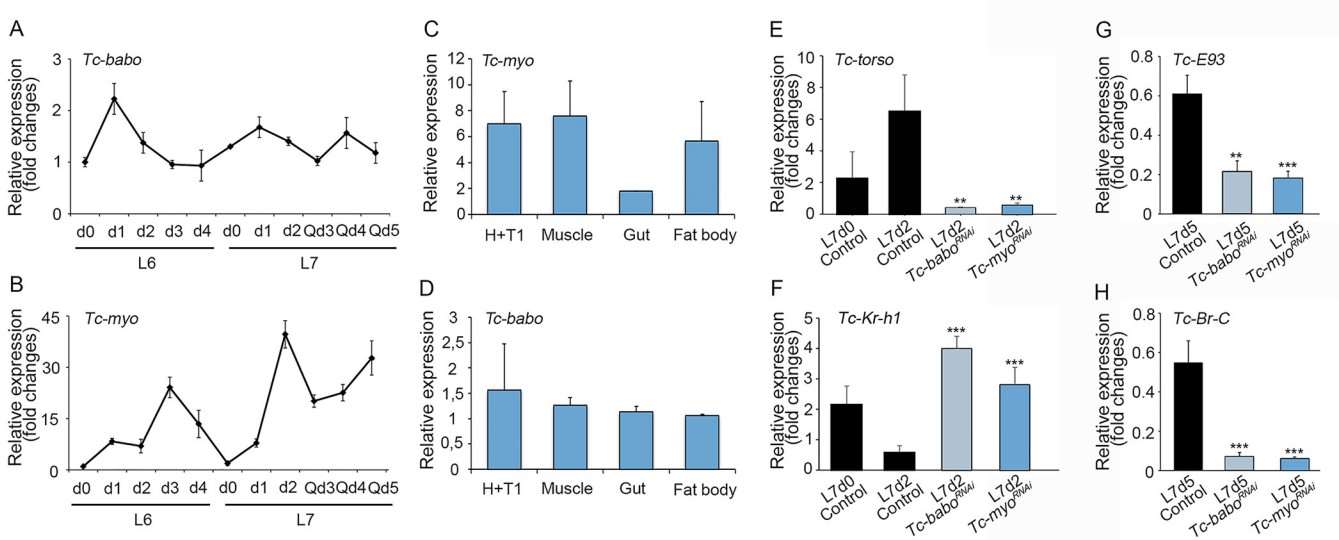

**Fig 5. TGFß/Activin signaling pathway activates *Tc-torso* expression through the repression of JH synthesis.** (A-B) Temporal changes in *Tc-babo* (A) and *Tc-myo* (B) mRNA levels measured by qRT-PCR in penultimate (L6) and ultimate (L7) instar larvae. Transcript abundance values were normalized against the *Tc-Rpl32* transcript. Error bars indicate the standard error of the mean (SEM) (n = 5). (C) *Tc-myo*, and (D) *Tc-trk* mRNA levels measured by qRT-PCR in head and first thoracic segment (H+T1), muscle, gut and fat body of day 2 L7 larvae. For transcript analysis, equal amounts of total RNA were used. Error bars indicate the SEM (n = 3). (E-H) Temporal changes in transcript levels of *Tc-torso* (E), *Tc-Kr-h1* (F), *Tc-E93* (G), and *Tc-Br-C* (H) measured by qRT-PCR at the indicated time points of L7 *Control*, *Tc-babo*$^{RNAi}$, and *Tc-myo*$^{RNAi}$ larvae. Transcript abundance values were normalized against the *Tc-Rpl32* transcript. Average values of three independent datasets are shown with standard errors (n = 5–8). Asterisks indicate statistically significant differences at **p < 0.01 and ***p < 0.001 (A t-test was used to compare the levels of gene expression for *Tc-babo*$^{RNAi}$, and *Tc-myo*$^{RNAi}$ with the L7d2 *control*). Raw data are in S1 Data (tab Fig 5).

receptor *Tc-babo* was consistently detected at similar levels in all the tissues measured, *Tc-myo*, in addition to muscle and fat body is primarily expressed in the head, where the PG is located and *Tc-torso* is mainly expressed (Fig 5C and 5D). To analyse the role of TGFß/Activin pathway during metamorphosis, we then depleted *Tc-babo* and *Tc-myo* during the last larval instar by injecting the corresponding dsRNAs (*Tc-babo*$^{RNAi}$ and *Tc-myo*$^{RNAi}$ animals). Remarkably, we found that in contrast to the normal upregulation of *Tc-torso* observed in *Control* larvae two days after the TS checkpoint, the levels of *Tc-torso* in *Tc-babo*$^{RNAi}$ and *Tc-myo*$^{RNAi}$ larvae were dramatically reduced, being even lower than the levels observed in newly molted L7 control animals (Fig 5E). In addition, *Tc-babo*$^{RNAi}$ and *Tc-myo*$^{RNAi}$ larvae presented persistently elevated levels of *TcKr-h1*, rather than the low levels observed in *Control* larvae (Fig 5F), which is indicative of sustained high levels of JH in these animals. Consistently, *Tc-E93* and *Tc-Br-C* expression did not increase in *Tc-babo*$^{RNAi}$ and *Tc-myo*$^{RNAi}$ larvae after reaching the TS (Fig 5G and 5H). Altogether these results show that the activity of TGFß/Activin signaling pathway at the onset of L7 is responsible for the decline in JH levels that triggers the changes in the expression of the temporal factors that control the metamorphic transition, including the sharp up-regulation of *Tc-torso*.

## The TGFß/Activin signaling pathway facilitates the metamorphic transition by regulating ecdysone levels

The results above indicate that *Tc-babo*$^{RNAi}$ and *Tc-myo*$^{RNAi}$ larvae do not initiate the molecular events that characterize the nature of the last larval instar, which involve the decline in *Tc-Kr-h1* expression with the concomitant upregulation of *Tc-E93*, *Tc-Br-C* and *Tc-torso*. It is expected, therefore, that *Tc-babo*$^{RNAi}$ and *Tc-myo*$^{RNAi}$ larvae would not initiate the larva-pupa

transformation at the ensuing molt. Consistent with this and in agreement with previous studies [31,32], *Tc-myo^RNAi* larvae failed to pupate and instead repeated successive larval molts to ultimately reach L10 instar larva (when we stopped analysing the knockdown animals). Remarkably, the intermolt period was significantly increased in these larvae (Fig 6A and 6B), and the weight of the supernumerary *Tc-myo^RNAi* larvae remained above the TS (Fig 6C), indicating that metamorphosis cannot be triggered in the absence of active TGFß/Activin signaling. In contrast, the phenotype of *Tc-babo^RNAi* larvae was even more dramatic as they never molted again and remained as L7 larvae an average of 65 days before dying (Fig 6D).

Given that *Tc-babo^RNAi* and *Tc-myo^RNAi* larvae exhibited consistently elevated levels of *Tc-Kr-h1* (Fig 5F), it raised the possibility that increased titers of JH were responsible for the lack of initiation of metamorphosis observed under these conditions. To validate this hypothesis, we inhibited JH synthesis by silencing *Tc-jhamt* in *Tc-babo^RNAi* and *Tc-myo^RNAi* larvae. Interestingly, the suppression of JH failed to promote metamorphosis in these larvae (Fig 7A), suggesting that the TGFß/Activin signaling pathway might also have a role in controlling the synthesis of ecdysone, as previously reported in *Drosophila* [29,31]. Supporting this possibility, the levels of the Halloween gene *Tc-phm* and the ecdysone-dependent gene *Tc-HR3*, used as proxies for 20E levels, were significantly downregulated in *Tc-babo^RNAi* and *Tc-myo^RNAi* larvae when compared to *Control* animals (Fig 7B and 7C), which demonstrate that TGFß/Activin signaling is required for ecdysone synthesis in *Tribolium*. Collectively, our findings reveal that TGFß/Activin signaling exerts a dual role in the control of metamorphic timing in *Tribolium*: firstly, it is responsible for the decline in JH levels at the TS checkpoint, triggering the genetic switch that initiate the metamorphic transition, including the up-regulation of *Tc-torso*; secondly, it controls the production of ecdysone, a critical factor required to elicit the corresponding developmental transition.

## Discussion

Here, we show that the Ptth-Trk/Torso signaling pathway plays a crucial role in the transition from juvenile to adult stages in *Tribolium*. Our data reveal that both Tc-Ptth and Tc-Trk ligands activate Tc-Torso signaling during the final larval stage of the beetle. However, whereas Tc-Ptth promotes ecdysone biosynthesis, Tc-Trk seems to induce growth. Interestingly, we found that the expression of *Tc-torso* depends on the TGFβ/Activin signaling-dependent decay of JH at the TS checkpoint. In addition, we also found that TGFβ/Activin signaling contributes to ecdysone biosynthesis. These findings improve our understanding of the molecular mechanisms underlying the hormonal control of metamorphic transitions and provide insights into the evolution of Torso signaling in insects.

### Postembryonic Torso signaling is activated by Tc-Ptth and Tc-Trk in *Tribolium*

Contrary to previously studied insects, our results demonstrated that post-embryonic Torso signaling in *Tribolium* is activated by both Tc-Ptth and Tc-Trk. To date, Ptth was the only ligand known to activate Torso during the post-embryonic stages in insects [14,36], whereas Trk was described as a Torso ligand exclusively involved in the generation of the terminal structures during the early stages of embryogenesis of *Drosophila* and *Tribolium* [25,26]. Despite the temporal specificity of the activation of the signaling by each ligand, it is important to note that both ligands seem to be functionally interchangeable, as ectopic expression of Ptth is able to activate Torso signaling in the early embryo of *Drosophila* [14], and overexpression of a processed form of Trk is able to induce precocious metamorphosis when overexpressed in the PG of the fly [26]. This indicates a possible common origin of both ligands, which have

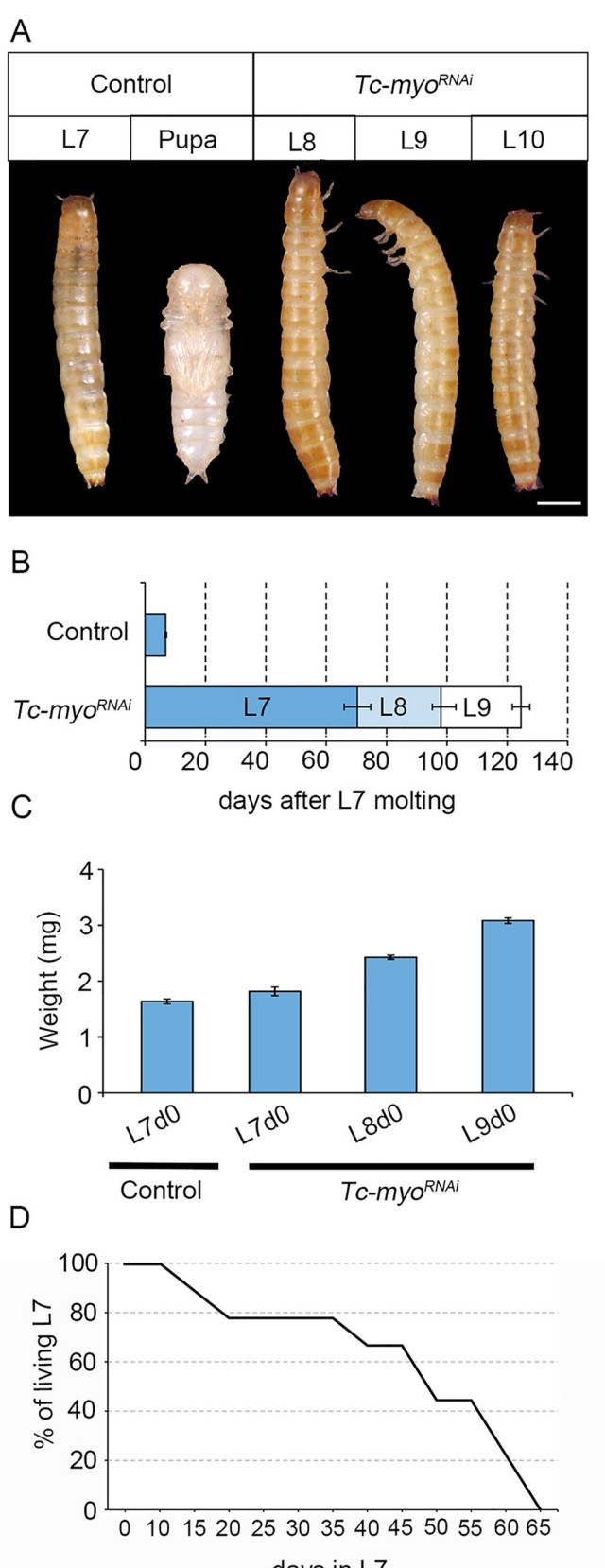

**Fig 6. Impairment of metamorphosis upon inactivation of TGFß/Activin signaling.** (A) Dorsal and ventral view of a Control L7 larva and pupa respectively, and dorsal views of supernumerary L8, L9, and L10 *Tc-torso*<sup>*RNAi*</sup> animals. Scale bar, 0.5 mm. (B) Temporal progression of L6 larvae injected with *dsMock* (Control) (n = 6) or with *dsTc-myo* (n = 15) and left until the ensuing molts. Each bar indicates the periods (mean ± SD) for each developmental stage after the molt into the L7 larval stage. (C) Growth in body weight of *Control* and *Tc-myo* animals. All weights are measured on day 0 of each instar. Bars indicate the mean ± SD (n = 8). (D) Percentage of *Tc-babo* depleted larvae alive recorded for each day after the molt into the L7 larval stage (n = 45). Raw data of B, C and D are in S1 Data (tab Fig 6).

acquired specific enhancers to activate the pathway in different tissues. The fact that Ptth appears to evolve from the original ligand Trk in the common ancestor of Hemiptera and Holometabola supports this idea [37].

Although we found that Tc-Trk and Tc-Ptth activates Torso signaling during the metamorphic transition in *Tribolium*, both ligands present common and non-overlapping functions. Whereas Tc-Ptth is involved in the regulation of ecdysone production and the timely induction of the metamorphic transition, Tc-Trk regulates the systemic growth during the last larval stage. Interestingly these data are consistent with results obtained in *Drosophila* and *Bombyx*, where inactivation of Torso specifically in the PG delays the onset of pupariation, extending the larval growth period and increasing the final pupal size [13,14,38], whereas depletion of *Torso* specifically in the fat body produced smaller pupae with no effect on developmental timing [35]. Although the ligand responsible for Torso activation in the *Drosophila* fat body has not been identified, these results raise the possibility that Trk might activate Torso in the fat body or other tissues of *Tribolium* to regulate growth rate. The enriched expression of *Tc-trk* in the fat body and the gut strongly supports this possibility, especially considering that the gut

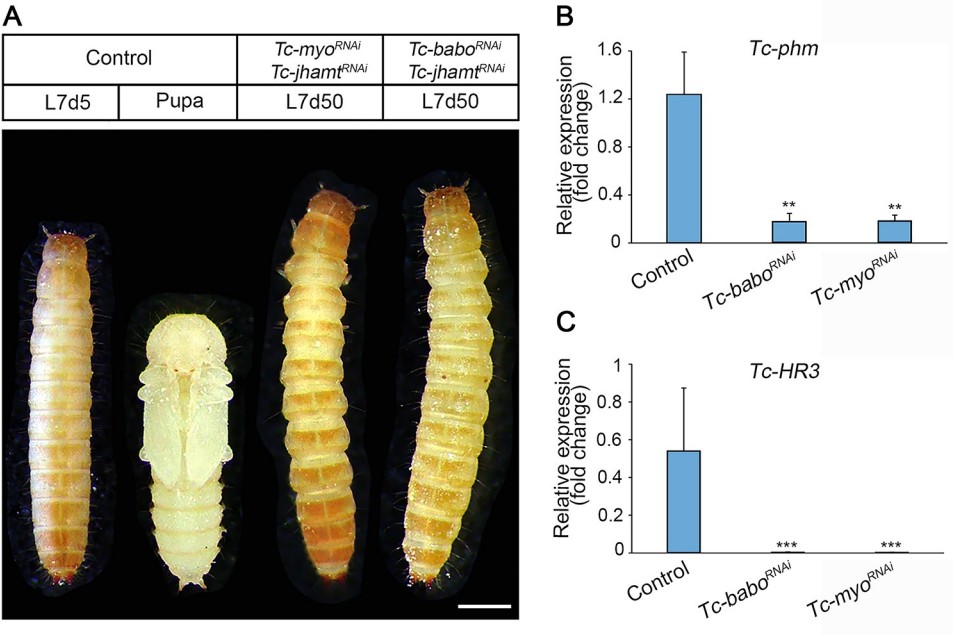

**Fig 7. Inactivation of TGFß/Activin signaling diminished ecdysone biosynthesis.** (A) Dorsal and ventral view of a Control L7 larva and pupa respectively, and dorsal views of double knockout *Tc-jhamt*<sup>*RNAi*</sup> and *Tc-myo*<sup>*RNAi*</sup>, and *Tc-jhamt*<sup>*RNAi*</sup> and *Tc-babo*<sup>*RNAi*</sup> L7 larvae, 50 days after injection. Scale bar, 0.5 mm. (B, C) Transcript levels of *Tc-phm*, (B) and *Tc-HR3* (C) measured by qRT-PCR in 2-day-old L7 Control, *Tc-babo*<sup>*RNAi*</sup> and *Tc-myo*<sup>*RNAi*</sup> larvae. Transcript abundance values were normalized against the *Tc-Rpl32* transcript. Average values of three independent datasets are shown with standard errors (n = 6). Asterisks indicate statistically significant differences at **p < 0.01 and ***p < 0.001 (t-test). Raw data of B and C are in S1 Data (tab Fig 7).

serves as a significant sensor of nutrients and plays a crucial role in transmitting food-related signals to the brain and other tissues [39]. However, it is also possible that Tc-Trk secreted from the gut and the fat body activates Tc-Torso in the PG to ensure an adequate supply of ecdysone for promoting normal growth. Generation of new genetic tools to deplete gene expression in a tissue-specific manner would be required to solve this question in *Tribolium*.

In *Drosophila* and in *Bombyx*, *torso* mutants produced larger adults due to a prolonged larval development that allows for extra growth [13,14,38]. However, our study shows that depletion of *Tc-Torso* in *Tribolium* reduces final pupal size even when the larval growth period is extended. This finding suggests that the activation of Tc-Torso by Tc-Trk in the fat body may play a crucial role in increasing larval body mass. Considering that Trk and Torso are the most ancient molecules of the signaling pathway, as indicated by their presence in chelicerates, the most basal group of arthropods [37], it is tempting to speculate that systemic growth regulation is likely the ancestral function of the pathway. The fact that co-option of Torso signaling in early embryogenesis as well as for ecdysone biosynthesis during post-embryonic development seems to be a relative evolutionary novelty in insect evolution support this idea [27,37]. Studying the function of Trk-Torso in hemipteran species and Trk-like molecules in chelicerates could shed light on this possibility.

## Up-regulation of *Tc-torso* depends on the TS checkpoint

Our expression analysis of *Tc-torso* in *Tribolium* reveals a peak of transcription at the TS checkpoint. This peak coincides with the attainment of the TS at the beginning of the final larval stage, accompanied by a decline in JH levels. The decrease in JH levels enables the up-regulation of Tc-E93 and Tc-Br-C, initiating the metamorphic transition. Interestingly, during the last larval stage an important increase of ecdysone production is required to induce the quiescent stage and the subsequent pupa formation [40]. In this sense, the decrease of JH production has been related to the upregulation of the Halloween genes, responsible for ecdysone biosynthesis. Thus, in *Drosophila*, inactivation of JH signalling in the PG triggers premature metamorphosis by de-repression of Halloween genes [41,42]. Such effect of JH on ecdysone biosynthesis might depend on the precocious up-regulation of *Torso* expression since ecdysone production relays in part on the activation of Torso signalling. Indeed, our results show that application of JH in the last larval stage of *Tribolium* impairs *Tc-torso* up-regulation. This effect has been also observed in *Bombyx* where the application of a JH analogue not only downregulates *Bm-torso* but also several Halloween genes [42,43]. These observations, therefore supports the idea that JH mainly represses ecdysone production by regulating *torso* expression.

## TGFβ/Activin signaling regulates JH biosynthesis

If the induction of *torso* expression depends on the disappearance of JH in the last larval stage, then what regulates such decline? Interestingly, in agreement with a previous report [32] we found that inhibition of JH synthesis depends on the activation of TGFβ/Activin signaling, as depletion of either its ligand *Tc-myo* or the main receptor *Tc-babo* blocks the metamorphic transition. Consistently, blocking the TGFβ/Activin signaling leads to a sustained high levels of the JH-dependent factor *Tc-Kr-h1*, indicating that JH levels do not decline under these circumstances. Similar results have been reported in other insects, suggesting a conserved role of TGFβ/Activin signaling in JH regulation. Thus, in *H. vigintioctopunctata*, *G. bimaculatus* and *B. germanica* activation of TGFβ/Activin signaling blocks JH production by directly downregulating of the JH biosynthetic enzyme gene *jhamt* [31,32,44]. The fact that JH regulation by TGFβ/Activin signaling occurs specifically during the last larval stage suggests a potential

relationship between body mass and the activation of TGFβ/Activin signaling. Interestingly, in the lepidopteran *Manduca*, increasing levels of *Ms-myo*, produced by the muscle, have been associated with the initiation of metamorphosis [32]. Similarly, we observed high levels of *Tc-myo* in the muscles of *Tribolium* during the last larval stage (Fig 5C). This observation strongly suggests that muscle growth during larval development induces a gradual increase in *Tc-myo* expression, activating the TGFβ/Activin signaling pathway once a certain threshold level is reached. Consequently, this activation leads to a decrease of JH production, allowing for the up-regulation of *Torso* and the subsequent rise of ecdysone levels. Such complex regulation might explain why Torso is only required during the last larval stage.

## Does TGFβ/Activin signaling act as a PG cell survival factor?

In addition to its role in regulating JH, TGFβ/Activin signaling also plays a role in ecdysone production in *Tribolium*, as revealed by *Tc-myo* knockdown. In agreement with a previous report [32], we found that depletion of *Tc-myo* in *Tribolium*, increased dramatically the inter-molt period and even induced a developmental arrest at L7 when the TGFβ/Activin receptor *Tc-babo* was knocked down. This phenotype is independent of the consistently high JH signaling activation detected under these conditions, as simultaneously blocking of both TGFβ/Activin and JH production failed to trigger metamorphosis (Fig 7). These results indicate that TGFβ/Activin signaling also plays a role in regulating ecdysone biosynthesis in *Tribolium*. In this context, the *Tc-babo* phenotype resembles the developmental arrest observed in *Drosophila* larvae when TGFβ/Activin signaling is inactivated in the PG [29]. As in *Drosophila*, arrested *Tc-babo*-depleted *Tribolium* larvae presented very low levels of *Tc-torso* expression with the consequent failure to induce the large rise in ecdysteroid titer that triggers metamorphosis (Fig 5E). Under these conditions, we anticipated the presence of supernumerary larvae in the subsequent molt due to the failure of JH decay. However, we found that inactivation of TGFβ/Activin signaling in *Tribolium* not only halts metamorphosis but also prevents larva molting, as depletion of *Tc-babo* leads to developmental arrest at L7 for more than 60 days. Considering that the effect of the injected dsRNAs is transient [45], the most likely explanation for the arrested *Tc-babo*-depleted larvae is that TGFβ/Activin might be required for PG viability during the last larval stage. This hypothesis is supported by the fact that inactivation of TGFβ/Activin signaling in adult *Drosophila* muscles reduces lifespan by modulating protein homeostasis in the tissue [46]. Likewise, reduction of TGFβ/Activin signaling in the cardiac muscle of *Drosophila* increases the cardiac autophagic activity [47]. Since autophagy is the main mechanism for larval tissue degradation during metamorphosis [48], it is tempting to speculate that TGFβ/Activin activation in the PG suppresses autophagy during the last larval stage of *Tribolium*. Alternatively, it is also plausible that inhibiting TGFβ/Activin signaling somehow sustains low ecdysone levels without affecting the viability of the gland. This could result in ecdysone levels remaining consistently insufficient to induce molting throughout the entire period. The characterization of the PG in *Tribolium* and specific analysis of the effects on TGFβ/Activin signaling in this tissue will provide further insights into this question.

In summary, our study reveals that the activation of Tc-Torso by two ligands, Tc-Ptth and Tc-Trk, during the last larval stage of *Tribolium* regulates the ecdysone-dependent timely transition to the metamorphic period and controls growth during this stage, thereby determining the final size of the insect. Furthermore, we showed *Tc-torso* up-regulation depends on larvae attaining the TS checkpoint at the onset of the last larval instar and that its expression depends on the decay of JH induced by the TGFβ/Activin signaling pathway. Our findings, therefore contribute to the understanding of how the coordination of different signaling pathways regulates the endocrine systems that control developmental growth and the timing of maturation in insects.

## Materials and methods

### Tribolium castaneum

The enhancer-trap line pu11 of *Tribolium* (obtained from Y. Tomoyasu, Miami University, Oxford, OH) was reared on an organic wheat flour diet supplemented with 5% nutritional yeast and maintained at a constant temperature of 29˚C in complete darkness.

### Quantitative Real-Time Reverse Transcriptase Polymerase Chain Reaction (qRT-PCR)

Total RNA from individual *Tribolium* larvae was extracted using the Mammalian Total RNA kit (Sigma). cDNA synthesis was performed following the previously described methods [49,50]. For quantitative real-time PCR (qPCR), Power SYBR Green PCR Mastermix (Applied Biosystems) was used to determine relative transcript levels. To standardize the qPCR inputs, a master mix containing Power SYBR Green PCR Mastermix and forward and reverse primers was prepared, with each primer at a final concentration of 100 μM. The qPCR experiments were conducted with an equal quantity of tissue equivalent input for all treatments, and each sample was run in duplicate using 2 μl of cDNA per reaction. As a reference, the same cDNAs were subjected to qRT-PCR using a primer pair specific for *Tribolium* Ribosomal *Tc-Rpl32*. All samples were analyzed on the iCycler iQReal Time PCR Detection System (Bio-Rad).

### Primer sequences used for qPCR for *Tribolium*

*Tc-torso-F*: 5′- TTGACGAGGAGAAGCTTCCAGAGT-3′
 *Tc-torso-R*: 5′- TGCAAATTGTTGCTGCATGTTGGT-3′
 *Tc-Ptth-F*: 5′-TCGTGTGGGATCGAATTTCGCGTTC-3′
 *Tc-Ptth-R*: 5′-GTCTTTCTGTTTCAAAACGCGGAC-3′
 *Tc-trk-F*: 5′-TATCGAGCAACTCGACGAT-3′
 *Tc-trk-R*: 5′-TCGATTTGCAATGCCACTGTT-3′
 *Tc-myo-F*: 5′-CAAGAAGTGCTCACCTTTGC-3′
 *Tc-myo-R*: 5′-CCTTCATGTACACGTACAG-3′
 *Tc-babo-F*: 5′-ATGGTGCATGGCTTCTGGTT-3′
 *Tc-babo-R*: 5′-AGTAAGTCGATGTAAAGCAGTA-3′
 *Tc-E75-F*: 5′-CGGTCCTCAATGGAAGAAAA-3′
 *Tc-E75-R*: 5′-TGTGTGGTTTGTAGGCTTCG-3′
 *Tc-phm-F*: 5′-TGAACAAATCGCAATGGTGCCATA-3′
 *Tc-phm-R*: 5′-TCATGGTACCTGGTGGTGGAACCTTAT-3′
 *Tc-Kr-h1-F*: 5′-AATCCTCCTGCTCATCCAGCACTA-3′
 *Tc-Kr-h1-R*: 5′-CAGGATTCGAACTAGGAGGTGTTA-3′
 *Tc-E93-F*: 5′-CTCTCGAAAACTCGGTTCTAAACA-3′
 *Tc-E93-R*: 5′-TTTGGGTTTGGGTGCTGCCGAATT-3′
 *Tc-Br-C-F*: 5′-TCGTTTCTCAAGACGGCTGAAGTG-3′
 *Tc-Br-C-R*: 5′-CTCCACTAACTTCTCGGTGAAGCT-3′
 *Tc-Rpl32-F*: 5′-CAGGCACCAGTCTGACCGTTATG-3′
 *Tc-Rpl32-R*: 5′-CATGTGCTTCGTTTTGGCATTGGA-3′

### Larva RNAi injection

*Tc-torso* dsRNA (IB_04720), *Tc-Ptth* dsRNA (IB_09326), *Tc-trk* dsRNA (IB_06187), *Tc-myo* dsRNA (IB_05899), *Tc-jhamt* dsRNA (IB_04499), and *Tc-babo* dsRNA (IB_03525) were synthesized by Eupheria Biotech Company. The control dsRNA used a non-coding sequence

from the pSTBlue-1 vector (dsMock). For larval injections, 1 μg of dsRNA was administered to penultimate instar and antepenultimate instar larvae. In cases of co-injection of two dsRNAs, equal volumes of each dsRNA solution were mixed and applied in a single injection.

### Nutritional experiments

*Tribolium* pupae-larvae were reared on a normal diet consisting of organic wheat flour containing 5% nutritional yeast. For the starvation experiment, newly molted L7 larvae, raised on the normal diet, were transferred to a new plate without any food. After 2 days, the larvae were collected for *Tc-torso* expression measurement using RT-qPCR.

### Treatment with Methoprene

To conduct the juvenile hormone mimic treatment, white L7 newly molted larvae of *Tribolium* were topically treated on their dorsal side with 1μg of isopropyl (E,E)-(RS)-11-methoxy-3,7,11-trimethyldodeca-2,4-dienoate per specimen in 1 μL of acetone. The control group received the same volume of solvent. At the desired stage, the larvae were subjected to mRNA expression analysis.

### Microscopy analysis

All pictures were obtained with AxioImager.Z1 (ApoTome 213 System, Zeiss) microscope, and images were subsequently processed using Fuji and Adobe photoshop.

### Supporting information

**S1 Data. Raw data and statistics summary of Figures.**
(XLSX)

### Author Contributions

**Conceptualization:** David Martín, Xavier Franch-Marro.

**Data curation:** Sílvia Chafino, Josefa Cruz, David Martín, Xavier Franch-Marro.

**Formal analysis:** Josefa Cruz, David Martín, Xavier Franch-Marro.

**Funding acquisition:** David Martín, Xavier Franch-Marro.

**Investigation:** Sílvia Chafino, Roser Salvia, Josefa Cruz, David Martín, Xavier Franch-Marro.

**Methodology:** Roser Salvia, Josefa Cruz, David Martín, Xavier Franch-Marro.

**Project administration:** David Martín, Xavier Franch-Marro.

**Resources:** David Martín, Xavier Franch-Marro.

**Supervision:** David Martín, Xavier Franch-Marro.

**Validation:** Josefa Cruz, David Martín, Xavier Franch-Marro.

**Visualization:** David Martín, Xavier Franch-Marro.

**Writing – original draft:** Xavier Franch-Marro.

**Writing – review & editing:** Sílvia Chafino, Josefa Cruz, David Martín, Xavier Franch-Marro.

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
