## [Decision Letter · Decision Letter 0]

24 Aug 2023

Dear Dr Franch-Marro,

Thank you very much for submitting your Research Article entitled 'TGFß/activin-dependent activation of Torso controls the timing of the metamorphic transition in the red flour beetle Tribolium castaneum' to PLOS Genetics.

The manuscript was fully evaluated at the editorial level and by independent peer reviewers. The reviewers appreciated the attention to an important topic but identified some concerns that we ask you address in a revised manuscript.

We therefore ask you to modify the manuscript according to the review recommendations. Your revisions should address the specific points made by each reviewer.

Yours sincerely,

Lynn M Riddiford

Academic Editor

PLOS Genetics

Gregory P. Copenhaver

Editor-in-Chief

PLOS Genetics

In making your revisions, please pay particular attention to the additional experiments suggested by Reviewer 1.

Reviewer's Responses to Questions

**Comments to the Authors:**

Reviewer #1: In the current manuscript, Chafino and others describe a signaling network that controls the timing of metamorphosis in the red flour beetle Tribolium castaneum. The authors present compelling evidence suggesting that the metamorphosis timing is differentially regulated by two ligands of the receptor tyrosine kinase named Torso, whose expression is in turn induced by the decline of juvenile hormone (JH). They also demonstrate that JH production in the last instar Tribolium is suppressed by a TGF-beta ligand named Myoglianin. By suppressing JH signaling and thereby promoting Torso expression, Myoglianin functions as a critical initiator of metamorphosis. Their model is consistent with the fact that Myoglianin initiates metamorphosis in many insect species, and it provides an important framework for understanding this critical process of insect development.

Overall, experiments are carefully designed and thoroughly conducted, making their proposed model convincing. Although there is some remaining uncertainty regarding specific functions of some genes such as Tc-torso, it requires development of new genetic tools for them to fully address those remaining issues. Having said that, I would like to propose two additional experiments that are still doable with available molecular tools, which I believe will significantly strengthen their argument. They are described below as two major comments:

Major comments:

1) Although their claim that Tc-Torso has two ligands in the last instar Tribolium is exciting, currently their claim is not convincing enough, as they do not provide any data that can explain how two ligands of a single receptor can have different functions. As the authors propose differential functions of Torso signaling in two different tissues (the PG and fat body), it would be interesting to check tissue specific expression patterns of Torso, as well as its two ligands, PTTH and Trunk. I understand that the PG is not identified in Tribolium yet, but the authors can still check the expression levels of these genes in the CNS and fat body, as well as some other tissues. I guess PTTH is specifically expressed in the CNS, but how about Trunk? Is it expressed in peripheral tissues, such as the fat body? Is Tc-torso really expressed in the fat body? This is a simple and doable experiment, whose outcome will be useful to interpret some of the presented data.

2) Currently, it is unclear whether (and to what extent) the low ecdysone signaling in babo-RNAi and myo-RNAi L7 larvae can be explained by the high JH titer in these animals on the PG. Although suppression of torso expression by JH alone cannot explain the strong arrest phenotype of these RNAi larvae, it is possible that JH can suppress ecdysone production in the PG in multiple ways. It is therefore very interesting to try double knockdown of babo and jhamt, as well as myo and jhamt in L7 larvae. The results of this experiment would tell us whether myo-babo signaling directly promotes ecdysone production in the PG, or the effect is rather indirect through its suppression of the JH titer.

Minor comments:

3) p. 5, line 75: “increased” should be “increase.”

4) p. 5, line 77: “Myostatin” should be “Myoglianin.”

5) p. 6, lines 122-123, “(Tc-Ptth RNAi animals) induced a pupariation delay of 4 days, compared to the 7 days observed in Tc-torso RNAi larvae”: I am not sure if this statement is accurate based on the data shown in Figure 2A. It appears that the difference of their median values is less than two days. Also, please label the Y axis of Figure 2A with tick marks, so we can read the data more accurately.

6) p. 7, line 140: “puparium” should be either “pupation” or “pupariation.” Also, it seems that these two words are mixed throughout the manuscript (e.g. the Y axis of Figure 2A is labeled “pupariation”, but the authors say that larvae “pupated,” for example on p. 6, line 113. This wording needs to be standardized.

7) p. 8, line 186: The word “Torso” is missing.

8) p. 10, line 231: “Figure 5E and E” should be “Figure 5E and F.”

9) p. 13, lines 322-324, the sentence “Thus, … jhamt [31, 32, 42]”: I guess “inactivation of TGFbeta/Activin signaling” here should be just “TGFbeta/Activin signaling.” Otherwise, it is the opposite of what they have shown in their current study. Also, “directly downregulation of” should be “directly downregulating.”

Reviewer #2: The manuscript by Chafino et al., investigates the role of the PTTH, Trunk and Myo signaling molecules in regulating entry of T castaneum larva into metamorphosis. PTTH and Trk are the two know ligands for the receptor tyrosine kinase receptor Torso, while Myo is one of the known TGF beta /Activin-like ligands that signal through the type I receptor Babo. Both Torso and Babo signaling has been implicated previously in regulating the metamorphic transition in several holometabolous insects, but details of molecular interplay between these signaling pathways has remained elusive. In this manuscript, the authors use gene expression levels and RNAi knockdown methods to tease apart the interactions between these two signaling pathways that control initiation of metamorphosis in Tribolium. They two major new discoveries. The first is that, unlike Drosophila where Trk and PTTH have very stage specific and non-overlapping functions during embryogenesis and metamorphosis respectively, In Tribolium Trk also activates Torso during the larval phase. Specifically, it appears to specifically regulate larval growth during the last larval stage while PTTH is responsible for triggering metamorphosis. This is a very intriguing discovery and the interpretation with regards to evo/dev mechanisms in the discussion is quite enlightening. The second and perhaps more important new mechanistic finding is that way in which the Myo/Babo pathway intersects with the PTTH/Trk pathway. The authors convincingly show that Myo /Babo signaling is responsible for shutting down JH synthesis at the end of the larval phase. This in turn stimulates high level expression of Torso so that, in response to PTTH, ecdysone biosynthesis is enhanced. This finding validates - at the molecular mechanism level- the long held textbook view that it is a combination of low JH and high E the stimulates the metamorphic transition in holometabolous insects. Overall, I find the experiments to be well done and the interpretations convincing. It is too bad that the genetics are not far enough along in Tribolium to look at the interaction in more detail such as what is the tissue source of the Myo (muscle??) and is Babo expressed and required in the CA for JH downregulation. However, even without these additional details, I think the paper contributes a major advance to our understanding of the signaling cascades that regulate production of the two key hormones that trigger metamorphosis in holometabolous insects.

Some minor comments

Since basal levels of Ecdysone production are required for imaginal disc growth in Drosophila especially during the last instar, is it possible that Tc -rk exerts its effects on the L7 growth via the basal level of ecdysone production and not via an effect of Trk signaling through Torso in another tissue (fatbody) as suggested in the discussion? What is the phenotype of knocking down an E biosynthetic enzyme in the final instar at different times. If one knocks down early is the body size (growth rate) affected differentially compared to knockdown in the last couple of days of L7?

Is it possible to inject Torso mRNA into starved larva to see if it can bypass the block in attaining the TS threshold?

If the function of Babo/Myo signaling is primarily to downregulate JH signaling in the last instar, one might predict that double knockdown of babo and Tc-Kr-h1 might allow normal molting. Have the authors tried his experiment?

Minor comments:

Page 6 line 114, should read Dm-PTTH and Dm-Torso not Tc-Ptth and Tc-torso.

Page 6 115-116 the sentence starting “These results suggest …. Is awkward using the parenthesis. It would read better to express as two separate thoughts in two sentences.

Page 8 line 186 TGFbeta/Activin signaling pathway regulates (insert Torso) expression…

Page 13 line 340 …knocked down…

Reviewer #3: In this study, Chafino et al explore the molecular regulation of the threshold size. They convincingly demonstrate that Torso expression is upregulated upon threshold size attainment and that it is only required during the final instar. Furthermore, they show that Torso mediates the effect of PTTH on Halloween gene expression and Trk on final instar growth. Finally, they demonstrate that the Torso expression is regulated by TGF-b/Activin signaling. The identification of Torso as a final instar-specific marker for the attainment of threshold size is novel and interesting. The study is well designed, and the data are clearly presented. I do have several issues that I would like the authors will address.

Major comments

- The referenced figures do not seem to point to the correct figures in some places. For example, in line 115, Fig 2C-E are listed as showing the smaller body size of Torso RNAi pupae. This should just be Figs 2D and E. In addition, in line 121, Fig. 2C is listed as showing the expression of PTTH; however, in the figure, PTTH is actually shown in Fig 2B. In line 173, I believe Fig. 3C should be cited instead of Fig. 3B. Please double check to make sure the referenced figures are accurate.

- The expression profile of trk is interesting as it exhibits a dramatic drop on day 4. Is there a significance here? For example, does it correspond to prepupal entry and cessation of feeding (similar to how insulin might respond to nutritional inputs)? If possible, it would be nice to have prepupal stage indicated in the expression profiles.

- Line 176-179: When jhamt3 and torso are simultaneously knocked down, the developmental delay increases from 5 days to 7 days. This is modest compared to the developmental delay seen in the final instar when torso alone is depleted (5->10 days). Given this, it might strengthen your arguments if you demonstrated that jhamt3 knockdown during L5 leads to precocious upregulation of torso in L6. I think such an experiment would be useful in demonstrating that torso is always upregulated in the instar prior to pupation. I don’t think this is absolutely necessary for publication, however, so I will leave this at the authors’ discretion.

- Line 300-306: The role of JH in Drosophila larvae is minimal (see Riddiford and Ashburner 1991, Gen. Comp. Endocrin. 82:172-183.). Given this, I would be cautious about making comparisons with Drosophila when it pertains to JH’s regulation of metamorphic onset. In many other insects, there is a “critical weight” after the “threshold weight” that is associated with the complete removal of JH from the hemolymph and hence entry into the prepupal stage. In these insects, PTTH can only be secreted after the critical weight has been reached (Nijhout and Williams, 1974, J Exp Biol 61: 493–501). I think the regulation of metamorphic transition in Tribolium is likely to be more similar to lepidopterans than it would be to Drosophila. I do agree that JH does appear to suppress Torso and the BASAL ecdysteroid biosynthesis in the final instar, and this effect is likely similar to what is observed in Bombyx (i.e. lines 307-311).

- The above comment brings up another issue, which is that in lepidopterans (and in Drosophila), the activation of Halloween genes is associated with the attainment of the minimum viable weight, which is reached relatively early after insect start feeding in the final instar and is attained after the threshold size is reached (see for example, Xu et al 2020. Insect Biochem Mol Biol 119, 103335). I therefore wonder if torso is upregulated at the threshold size or at the minimum viable weight? I just want to make sure the equivalent stages are named appropriately to avoid confusion in the future.

- Line 351-367: Here the authors explain the inability of TcBabo RNAi larvae to molt by proposing that the PGs might degenerate through autophagy. Specifically, they discuss the effect of inactivation of TGF-beta signaling on adult Drosophila cardiac muscles and suggest that TcMyo may be needed for PG survival. While possible, considering the fact that the knockdown of Myo in Blattella leads to hyperproliferation of PG cells and an increase in the PG size (Kamsoi and Belles, 2019, FASEB J. 33, 3659–3669), I would tend to favor the idea that the Halloween gene expression remains so low that ecdysteroid levels never reaches a high enough level to trigger a molt. Consequently, I would suggest discussing these other possibilities and changing the subheading of this section.

- Fig. 4 legend states that t-tests were used to compare the mean expression changes. Please specify what the two treatment groups are since there are more than two different treatment groups shown and t-tests should only be used for comparisons between two treatment groups.

Minor comments

Middle of the abstract: “Our data reveals” should be “Our data reveal”

Line 10: “Holomeabolous:” should be “Holometabolous”

Line 77” The authors use “Myostatin” as the name of the ligand. Although TcMyo is a homolog of Myostatin, it is also a homolog of GDF11. Later on, the authors use the term Myoglianin (line 193). I would suggest that the authors stick with Myoglianin throughout the manuscript.

Line 117: “might not only depends” should be “might not only depend”

Line 120: Tribolium should be italicized

Line 181: “trough” should be “through”

Line 219: “when we stop analyzing the knockdown animals” – change to ““when we stopped analyzing the knockdown animals”

Line 223: “On the other hand” should follow “on the one hand”. Perhaps use “In contrast”?

Line 231: “Fig. 5 E and F” instead of “5E and E”?

Line 227-228: I would suggest including Blattella to this list as well (Kamsoi and Belles 2019).

Line 232: “Collectively”

Line 243: “Trk promotes organ growth”. Can you say this with such certainty? Depletion of Trk leads to smaller body size, so this may be due to differences in organ growth, but one cannot necessarily rule out lowered feeding rates, for example.

Line 270: “this data is consistent” should be “these data are consistent”

Line 323: “blocks JH production by directly downregulation of…”: Change to “blocks JH production by directly downregulating…”

Line 332: “rise” instead of “raise”

Line 339: “inter-molting period” should be “intermolt period”

Line 374: Italicize Tribolium

Line 414: Please give the exact amount of dsRNA injected.

Line 429, 601: Italicize Tribolium

Line 603, 619: Italicize Tc-Rpl32

Line 617: Instead of just (B), it should probably be (B, C)

Fig 3 legend: The legend for 3A describes t-tests and asterisks. However, it looks like an ANOVA was used instead and letters are provided. Please remove line 634.

Line 670: Instead of just (E), it should probably be (E, F)

Fig. 1, 2 legends: Please define pd0 in the expression profile.

**Have all data underlying the figures and results presented in the manuscript been provided?**

Reviewer #1: Yes

Reviewer #2: Yes

Reviewer #3: Yes

PLOS authors have the option to publish the peer review history of their article (what does this mean?). If published, this will include your full peer review and any attached files.

Reviewer #1: No

Reviewer #2: **Yes: **Michael B. O'Connor

Reviewer #3: No

---

## [Editor Report · Decision Letter 1]

10 Nov 2023

Dear Dr Franch-Marro,

We are pleased to inform you that your manuscript entitled "TGFß/activin-dependent activation of Torso controls the timing of the metamorphic transition in the red flour beetle Tribolium castaneum" has been editorially accepted for publication in PLOS Genetics. Congratulations!

There are a few minor errors that need correcting that can be taken care of as you prepare your final draft for the production team (the editorial team will not need to re-evaluate): 

    lines 35, 135:  Since this refers only to Drosophila, use "pupariation" rather than "pupation".

    lines 77, 252, 385: titers

    line 161:  [26] is the wrong reference here.

Yours sincerely,

Lynn M Riddiford

Academic Editor

PLOS Genetics

Gregory P. Copenhaver

Editor-in-Chief

PLOS Genetics

Comments from the reviewers (if applicable):

**Data Deposition**

http://datadryad.org/submit?journalID=pgenetics&manu=PGENETICS-D-23-00875R1

**Press Queries**

---

## [Editor Report · Acceptance letter]

22 Nov 2023

PGENETICS-D-23-00875R1 

TGFß/activin-dependent activation of Torso controls the timing of the metamorphic transition in the red flour beetle *Tribolium castaneum*

Dear Dr Franch-Marro, 

We are pleased to inform you that your manuscript entitled "TGFß/activin-dependent activation of Torso controls the timing of the metamorphic transition in the red flour beetle *Tribolium castaneum*" has been formally accepted for publication in PLOS Genetics! Your manuscript is now with our production department and you will be notified of the publication date in due course.

With kind regards,

Lilla Horvath

PLOS Genetics

On behalf of:
